# Recognition of Dorsal Hand Vein in Small-Scale Sample Database Based on Fusion of ResNet and HOG Feature

**Jindi Li, Kefeng Li \*, Guangyuan Zhang \*, Jiaqi Wang, Keming Li and Yumin Yang**

School of Information Science and Electric Engineering, Shandong Jiaotong University, Jinan 250000, China
\* Correspondence: 205073@sdjtu.edu.cn (K.L.); zhanggy@sdjtu.edu.cn (G.Z.)

**Abstract:** As artificial intelligence develops, deep learning algorithms are increasingly being used in the field of dorsal hand vein (DHV) recognition. However, deep learning has high requirements regarding the number of samples, and current DHV datasets have few images. To solve the above problems, we propose a method based on the fusion of ResNet and Histograms of Oriented Gradients (HOG) features, in which the shallow semantic information extracted by primary convolution and HOG features are fed into the residual structure of ResNet for full fusion and, finally, classification. By adding Gaussian noise, the North China University of Technology dataset, the Shandong University of Science and Technology dataset, and the Eastern Mediterranean University dataset are extended and fused to from a fused dataset. Our proposed method is applied to the above datasets, and the experimental results show that our proposed method achieves good recognition rates on each of the datasets. Importantly, we achieved a 93.47% recognition rate on the fused dataset, which was 2.31% and 26.08% higher than using ResNet and HOG alone.

**Keywords:** ResNet; HOG; feature fusion; DHV recognition

## 1. Introduction

Biometric identification refers to a technology that uses the physiological or behavioral features of the human body (such as fingerprint features [1], face features [2], gait features [3], signature handwriting [4], etc.) to achieve identity authentication. Compared with the traditional authentication systems based on passwords, tokens, and certificates, biometric authentication systems have many advantages [5–7], so more and more people begin to focus on the research of biometric identification. As a kind of biometric identification method, dorsal hand vein recognition is different from other biometric identification methods. It mainly uses infrared light to collect images of the back of the hand and uses this method to show the outline structure of veins [8] in order to realize the identification of an individuals' identity. An anatomy article [9] has demonstrated that the DHV has a unique structure during growth and development, which can characterize the individual to a certain extent. Therefore, the research on DHV recognition is of great significance in individual recognition.

Currently, DHV recognition research is primarily focused on a single database, which makes it easier to achieve better recognition results due to the use of similar acquisition equipment, subjects, and collection environment. DHV identification on a single database includes two methods, namely traditional features and deep learning methods. In 2019, Vairavel et al. [10] studied the recognition performance of three classical dense descriptors for DHV recognition, including Local Binary Pattern(LBP), HOG, and Weber local descriptor (WLD), and achieved good results on the Northern University of Technology(NCUT) [11] database. Liu et al. [12] proposed an improved biometric map matching method in 2020, which achieved a 98.09% recognition rate on the Xi'an Jiaotong University (XJTU) database. With the rise of artificial intelligence, some researchers have begun to use deep learning methods for DHV identification. In 2019, Wang et al. [13] used the

selective convolution feature (SCF) model and spatial pyramid pooling (SPP) to obtain a more robust feature representation of images and they achieved excellent recognition rates on the China University of Mining and Technology (CUMT) databases. In 2019, Zhong et al. [14] designed a Deep Hashing Network (DHN) for DHV identification and achieved good results.

We can see from the above research that the DHV recognition of a single database has achieved good results, whether using traditional methods or deep learning methods. In recent years, researchers have begun to focus on cross-device DHV identification. In 2019, Wang et al. [15] proposed an improved scale-invariant feature transform (SIFT) algorithm, which achieved a recognition rate of 88.5% on datasets acquired by different devices by improving the scale factor $\alpha$, extremum search neighborhood structure, and matching threshold R. In 2021, Wang et al. [16] proposed a two-stage coarse-to-fine matching method. First, the vein images to be matched are roughly matched in each category of the database, and then the SIFT method is used to extract the feature points of the vein images for fine matching. Such a method achieves good results on the cross-device DHV database.

Through the investigation and research on the cross-device DHV, most of the research on the cross-device DHV is based on the database of the same group of subjects and does not consider that the different subjects may have a certain impact on the experimental results. For different databases, there is not only the problem of different sampling equipment but also the diversity of subjects. Therefore, taking into account the differences between equipment and subjects is a major challenge in this field. In addition, in the cross-device DHV research, most researchers use traditional methods, but traditional methods are not robust to noise; if deep learning methods are used, they will face the problem of data volume. Given the above problems, this paper makes the following contributions:

(1) We designed a network framework that fused ResNet and HOG features, tested them on three different small-sample datasets and achieved good results.
(2) Aiming at the less researched cross-database DHV recognition, a fusion database containing three different datasets was established, and the proposed feature fusion method was applied to this database, achieving a high recognition rate and strong robustness.

## 2. Materials and Methods

### 2.1. Data Processing

#### 2.1.1. Dataset

The databases used in this paper are the dataset of the Shandong University of Science and Technology (SDUST) [17], the dataset of the Eastern Mediterranean University of Turkey (FYO) [18], the dataset of NCUT, and the fusion dataset (Fusion Dataset).

(1) SDUST Dataset

The dataset is a database of DHVs collected from the left and right hands of 63 males and 47 females using a commercial infrared device DF-300. The dataset contains 40 images of each subject, 20 for the left and right hands, for a total of 220 categories. The pictures in each category achieve image enhancement and data enhancement by changing brightness and random rotation, the size is 640 × 480 pixels, the horizontal and vertical resolutions are 96 dpi, and the format is jpg, as shown in Figure 1a.

(2) FYO Dataset

The FYO dataset was collected by a team from the Eastern Mediterranean University, which uses homemade equipment to collect data. The dataset collected images of the DHVs, palm veins, and wrist veins of the left and right hands of 160 volunteers (111 males and 49 females) twice, with a 10-min interval between the two acquisitions. The original data of the dataset contains data collected twice, each time there were 320 images of DHVs, 320 images of palm veins, and 320 images of wrist veins, and the images were all 800 × 600 color images, as shown in Figure 1b.

(3)　NCUT Dataset

This dataset builds a database of images of the backs of the hands of 102 people, including 50 males and 52 females. During the collection, the left and right hands are alternately collected, that is, after collecting a vein picture with the left hand, the right hand is placed at the collection site to collect one image and then an image of the left hand is collected. This alternating method ensures the difference between the same type of samples. Due to the differences in the distribution of veins in the left and right hands of each individual, the database can be considered a back-of-hand image library composed of 204 types of samples. When collecting, 10 pictures are taken from the back of each hand, the size is 640 × 480 pixels, the horizontal and vertical resolutions are 96 dpi, the grayscale is 256 levels, and the format is bmp, as shown in Figure 1c.

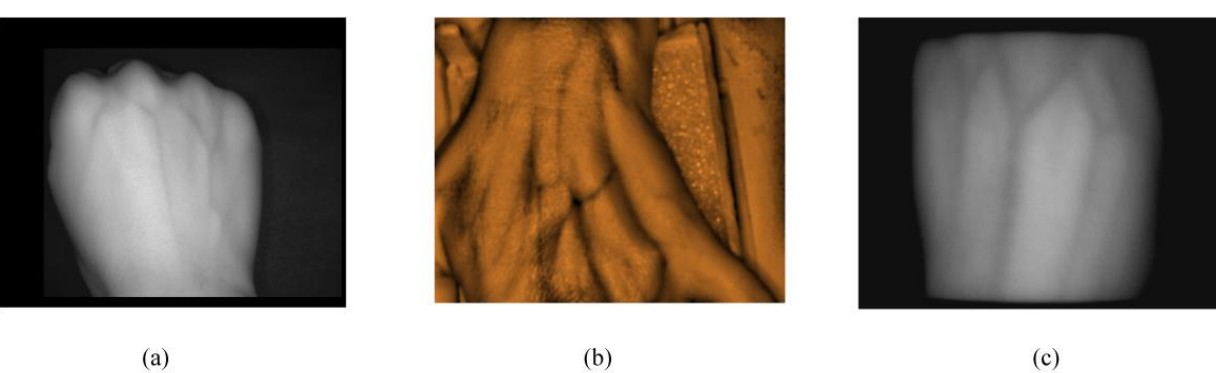

(a)　　　　　　　　　　　　　　　(b)　　　　　　　　　　　　　　　(c)

**Figure 1.** Datasets. (**a**) SDUST dataset. (**b**) FYO dataset. (**c**) NCUT dataset.

(4)　Fusion Dataset

The fusion dataset includes the Shandong University of Science and Technology dataset, Northern University of Technology dataset, and Turkey Eastern Mediterranean University dataset, with a total of 372 volunteers and 744 sample data. Since the size and format of the images in different datasets are different, the images must be preprocessed first. First the images in the FYO dataset were transformed to grayscale and then their size was normalized, and the normalized size is 640 × 480 pixels. In this way, the images of the fusion dataset are all grayscale images with a size of 640 × 480 pixels, as shown in Figure 2.

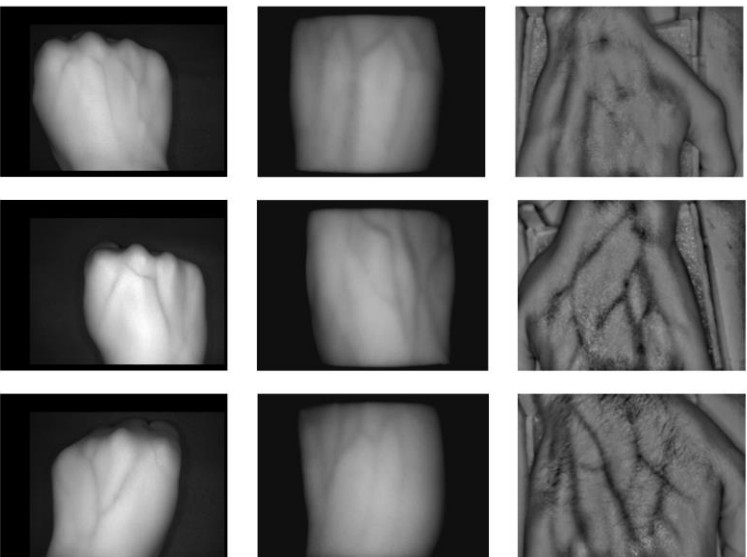

**Figure 2.** Fusion Dataset.

### 2.1.2. Extract Image ROI

As can be seen from Figures 1 and 2, there are large differences in the images of different DHV databases, including changes in rotation angle, size, brightness, and noise. This is mainly due to the differences in parameters, such as contrast, brightness, focal length, and optical performance of the lens between different acquisition devices, as well as the state of the collector's hand. These factors have a significant impact on the recognition results, and simple scale normalization is not conducive to extracting the texture features of the samples. Therefore, we need to extract the ROI of the image, and the method of extracting ROI is studied in [19,20]. In this paper, the centroid $(x_0, y_0)$ adaptive method is used to determine the ROI area of the DHV image. The centroid of the vein image expressed by $G(x, y)$ can be calculated as:

$$x_0 = \frac{\sum\limits_{i,j} i \times g(i,j)}{\sum\limits_{i,j} g(i,j)}; y_0 = \frac{\sum\limits_{i,j} j \times g(i,j)}{\sum\limits_{i,j} g(i,j)} \tag{1}$$

where $g(i,j)$ is the grayscale value of pixel $(i,j)$.

A square area with the size of R × R pixels is extracted and centered as the vein image to be processed. The experiment [21] verifies that when the ROI of the vein image is 380 × 380, the recognition rate can achieve the best effect, as shown in Figure 3.

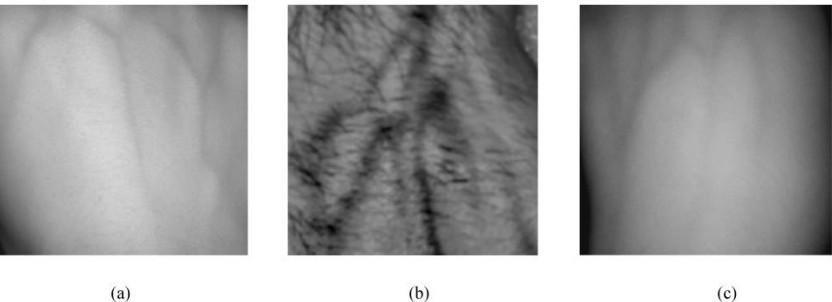

|     |     |     |
| :-: | :-: | :-: |
| (a) | (b) | (c) |

**Figure 3.** ROI images from different datasets. (**a**) ROI of SDUST dataset. (**b**) ROI of FYO dataset. (**c**) ROI of NCUT dataset.

### 2.1.3. Add Gaussian Noise

Gaussian noise is a kind of noise whose probability density function obeys normal distribution. The main function of Gaussian noise injection, as a data enhancement technique, is to add random Gaussian noise to samples to reduce overfitting during model training. Since there is only one image for each person in each dataset, it is not enough to prove the performance of the proposed method. Therefore, this method is adopted in this paper to expand the dorsal vein dataset. The dataset after adding Gaussian noise is shown in Figure 4.

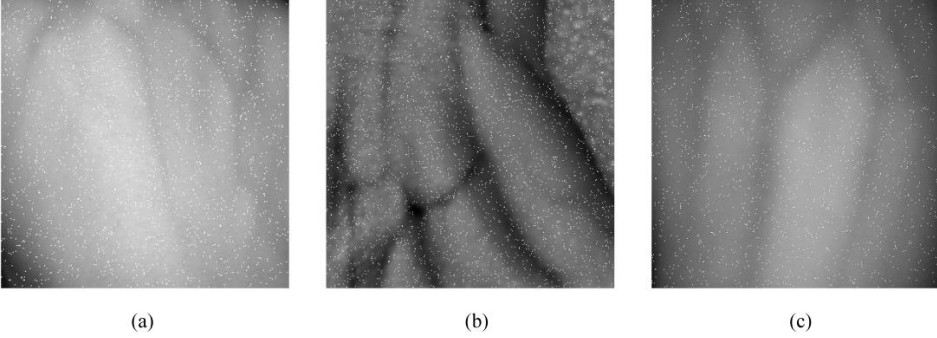

|     |     |     |
| :-: | :-: | :-: |
| (a) | (b) | (c) |

**Figure 4.** Images after adding Gaussian noise. (**a**) SDUST dataset after adding Gaussian noise. (**b**) FYO dataset after adding Gaussian noise. (**c**) NCUT dataset after adding Gaussian noise.

### 2.2. Related Algorithms

#### 2.2.1. HOG

HOG [22] feature is a feature descriptor used in computer vision and image processing for object detection, which constitutes a feature by computing and counting the gradient direction histograms of local regions of an image.

The acquisition of HOG features is divided into four steps:

The first step is to normalize the color space of the DHV image, which consists of two aspects, image grayscale, and Gamma correction. Because our image is already a grayscale map, only Gamma correction is performed, and the Gamma correction formula is as shown in Formula (2).

$$I(x,y) = I(x,y)^\gamma, (\gamma = 0.5) \tag{2}$$

The gradient is calculated in the horizontal and vertical directions in the second step, and the gradient calculation formula are shown in Formulas (3) and (4).

$$G_x(x,y) = H(x+1,y) - H(x-1,y) \tag{3}$$

$$G_y(x,y) = H(x,y+1) - H(x,y-1) \tag{4}$$

where $G_x(x,y)$,$G_y(x,y)$, and $H(x,y)$ denote the horizontal gradient, vertical gradient, and pixel value at pixel point $(x,y)$ in the input image, respectively. The amplitude and direction of the gradient at pixel $(x,y)$ are shown in Formulas (5) and (6).

$$G(x,y) = \sqrt{G_x(x,y)^2 + G_y(x,y)^2} \tag{5}$$

$$\alpha(x,y) = \tan^{-1}\left(\frac{G_y(x,y)}{G_x(x,y)}\right) \tag{6}$$

The third step is to divide the image into $8 \times 8$ pixel cells. As shown in the red grid in Figure 5, a total of 784 cells are included, and the feature descriptors of each cell are counted. There are 9 descriptors for each cell, representing from $0°$ to $160°$. Every 4 cells is a block, which is represented by the yellow grid in Figure 5. It contains a total of 729 blocks. The descriptors of all cells in each block are the HOG features of the block.

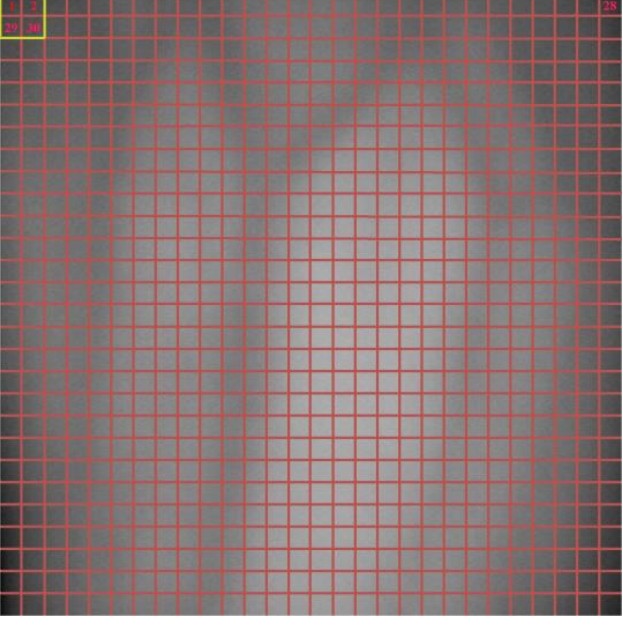

**Figure 5.** Divided into $28 \times 28$ cell images of veins.

The fourth step is to concatenate the HOG feature descriptors of all blocks in the image to represent the HOG feature of the image, which is a $1 \times 26{,}244$ vector.

### 2.2.2. ResNet Network

Before the idea of residual learning was proposed, traditional convolutional networks or fully connected networks had more or fewer problems, such as information loss and loss when information was transmitted. In addition, deep networks cannot be trained when gradients are small or exploding. ResNet [23] is a deep learning network that solves the problem of network degradation by introducing a deep residual learning framework. The network uses a residual unit structure, as shown in Figure 6. Assuming that the input feature is x, the learned feature is $H(x)$ and the residual unit of the learned feature can be represented as $F(x) = H(x) - x$. The equation of $F(x) + x$ can be implemented by a feedforward neural network with shortcut connections, and the residual unit structure can avoid the feature loss of the convolutional layer during the information transmission process.

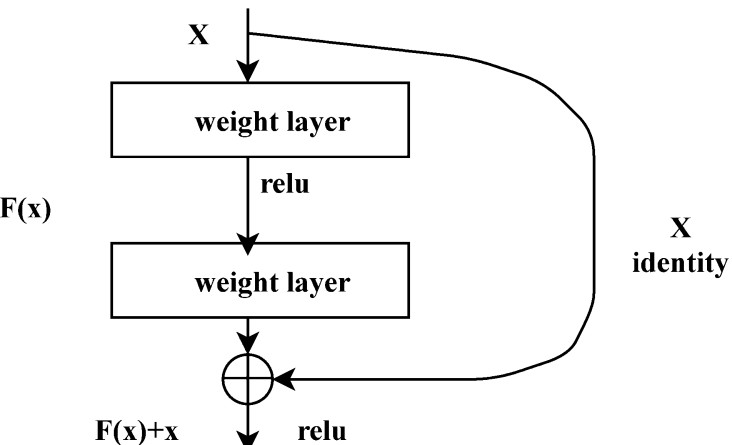

**Figure 6.** Residual structure.

Figure 7 shows the residual structure of ResNet34, Table 1 shows the network parameters for the ResNet34. The main branch of the residual structure is composed of two layers of $3 \times 3$ convolutional layers, and the connecting line on the right side of the residual structure is the shortcut branch, that is, the identity branch. Such branches are designed to reduce the amount of computation and parameters.

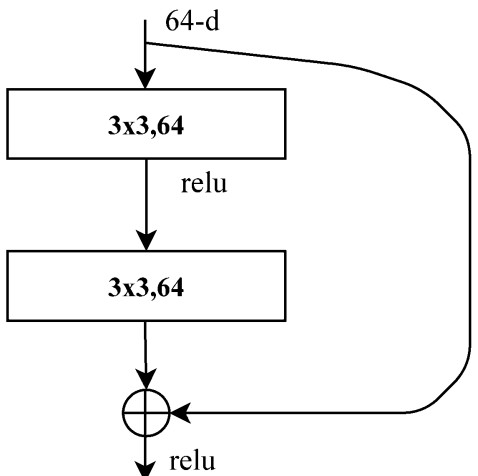

**Figure 7.** ResNet34 residual structure.

**Table 1.** The ResNet34 network parameters in this paper. (* represents the number of categories classified.).

| Layer Name | Network Parameters | Input Size | Output Size |
|---|---|---|---|
| conv1 | $7 \times 7, 64$, stride2 | $224 \times 224 \times 3$ | $56 \times 56 \times 3$ |
| Conv_block1 | $\begin{bmatrix} 3 \times 3, & 64 \\ 3 \times 3 & 64 \end{bmatrix} \times 3$, stride2 | $56 \times 56 \times 64$ | $56 \times 56 \times 64$ |
| Conv_block2 | $\begin{bmatrix} 3 \times 3, & 128 \\ 3 \times 3 & 128 \end{bmatrix} \times 4$, stride2 | $56 \times 56 \times 64$ | $28 \times 28 \times 128$ |
| Conv_block3 | $\begin{bmatrix} 3 \times 3, & 256 \\ 3 \times 3 & 256 \end{bmatrix} \times 6$, stride2 | $28 \times 28 \times 128$ | $14 \times 14 \times 256$ |
| Conv_block4 | $\begin{bmatrix} 3 \times 3, & 512 \\ 3 \times 3 & 512 \end{bmatrix} \times 3$, stride2 | $14 \times 14 \times 256$ | $7 \times 7 \times 512$ |
| AdaptiveAvgPool2d (H, W) | H = 1, W = 1 | $7 \times 7 \times 512$ | $1 \times 1 \times 512$ |
| FC | \ | 512 | * |

*2.3. Proposed Methods*

2.3.1. Fusion of ResNet and HOG Feature

The framework based on the fusion of ResNet and HOG features proposed in this paper is shown in Figure 8.

First, we do two-way processing on the image input to the neural network, in which we perform a convolution operation on it to extract the low-level semantic information of the image as a Feature Map. The other way is to input into the HOG function to extract the gradient information of the image. When performing HOG feature extraction on images, we make some changes to the features. First, we obtain the HOG feature of the entire image according to the general process, and the extracted feature vector is a one-dimensional vector of $1 \times 26{,}244$, as shown in Equation (7).

$$V = [l_1, l_2, \cdots, l_{26244}] \tag{7}$$

Since the acquired image HOG feature dimension is large, the feature vector needs to be normalized. Otherwise, the image features are jerkier in gradient descent and the model has difficulty converging when the neural network is learning. Therefore, the obtained one-dimensional vector is first normalized, and the normalization formula is shown in Formula (8).

$$f_{out} = \frac{x_i - \min(x)}{\max(x) - \min(x)} \tag{8}$$

After the normalization is completed, the normalized feature vector is reshaped into a feature map of $162 \times 162$, as shown in Formula (9), to obtain the HOG feature.

$$Feature\_HOG = \begin{bmatrix} l_1 & \cdots & l_{162} \\ \vdots & \ddots & \vdots \\ l_{26082} & \cdots & l_{26244} \end{bmatrix} \tag{9}$$

Since the size of HOG_Map is different from that of Feature_Map, a convolution operation is required. The convolved HOG feature is HOG_Feature, and then spatial feature fusion is performed with Feature_Map. The fusion method is shown in Equation (10), and the fusion method is shown in Formula (11). Then input the fused features into the ResNet residual block, and finally reduce the dimension of the feature map output by the ResNet residual block and input it into the fully connected layer for classification.

$$y_{c,h,w}^{sum} = \alpha x_{c,h,w}^a + (1 - \alpha) x_{c,h,w}^b \tag{10}$$

$$Feature\_Fusion\_Map = \alpha \times HOG\_Feature + (1 - \alpha) \times Feature\_Map \qquad (11)$$

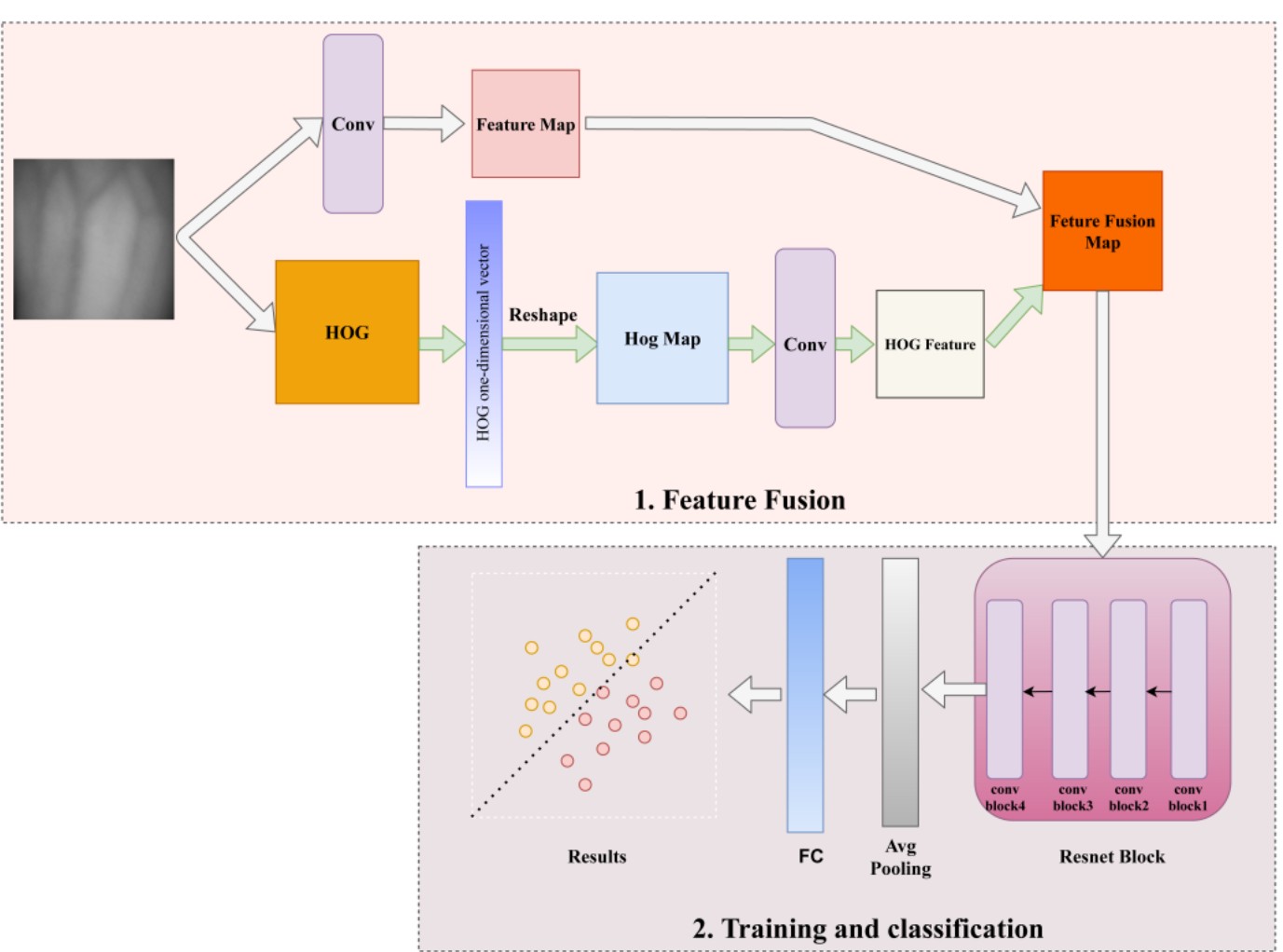

**Figure 8.** ResNet and HOG feature fusion methods. (1.) Feature Fusion. This part obtains the HOG feature and shallow semantic information of the image, respectively, and then performs spatial feature fusion. (2.) Training and classification. The features after feature fusion are input into the residual architecture of ResNet for training, then input into the average pooling layer for dimensionality reduction, and finally input into the fully connected layer for classification.

2.3.2. Feature Fusion Parameter Selection

When dividing the image into cells, we performed three sets of experiments on the Twenty dataset of NCUT to verify the effect of the number of cells on feature fusion. The experimental results are shown in Table 2.

**Table 2.** Different Cell identification results.

| Number of Cells | Evaluation Methods | |
|:---:|:---:|:---:|
| | Recognition Rate (%) | Train Time (s) |
| 196 | 85.94 | 250 |
| 256 | 86.12 | 267 |
| 784 | **86.53** | **330** |
| 3136 | 86.46 | 694 |

As can be seen from Table 2, as the number of cells increases from 196 to 784, the feature fusion recognition rate gradually increases. However, as the number of cells increases, when reaching 3136 cells, the recognition rate not only does not increase but instead decreases. We deduce that the reason is that when the number of cells increases to a certain level, if the number of cells is increased, the gradient information of each cell will be lost to a certain extent, resulting in a decrease in the recognition effect. It can also be seen from Table 2 that as the number of cells increases, the model training time also increases. Considering the above two factors, we chose the number of cells to be 784.

## 3. Experiments and Analysis

### 3.1. Feature Fusion Validity Experiments

In this paper, the dataset is divided into a training set, validation set, and test set according to 8:1:1 by random division. In addition, to ensure the persuasiveness of the experimental results, we conducted each experiment three times on the test set and took the average value. We used the Pytorch deep learning framework. The graphics card was NVIDIA GeForce RTX 2080 Ti 16 GB, the batch size was 16, the learning rate was 0.001, the loss function is the cross entropy loss function, and the epochs were 50.

Before feature fusion, we performed nine experiments on NCUT's Twenty dataset to find the best fusion factor $\alpha$, and the experimental results are shown in Figure 9. When the fusion factor is $\alpha = 0.3$, the recognition rate can achieve the best effect. Therefore, the fusion factors of ResNet and HOG in the following experiments are both set to 0.3.

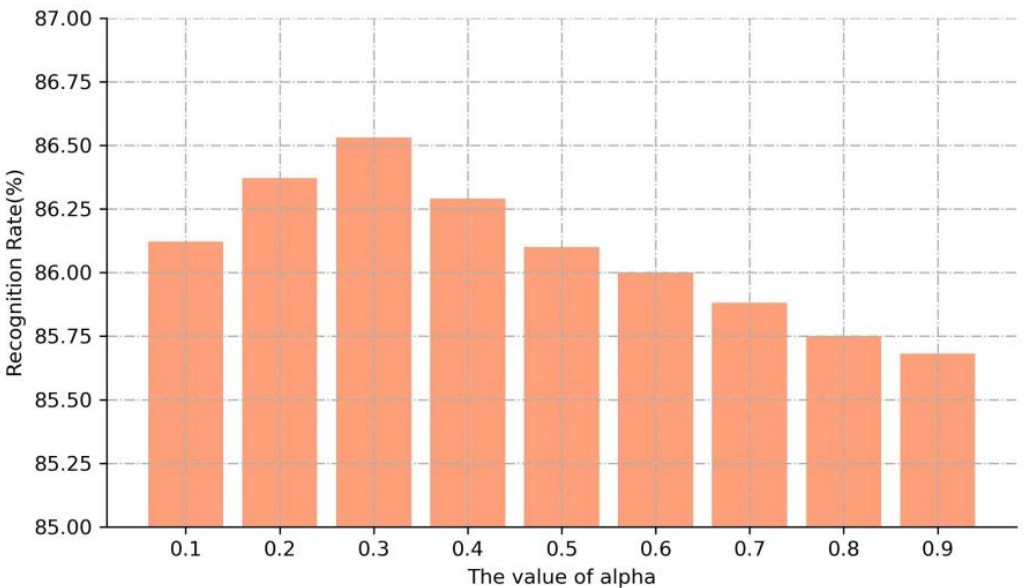

**Figure 9.** ResNet and HOG fusion factor take values.

Table 3 shows the recognition rates on a single dataset using ResNet, HOG, and ResNet_HOG methods. First of all, it can be seen from Table 3 that using the ResNet network can achieve a recognition rate of more than 90% on the Fifty dataset of a single database, but it does not achieve such a high effect on the Twenty dataset. This is because the larger the number of samples, the better the trained model will be, and the stronger the generalization ability of the model. Secondly, we can see from Table 3 that as the number of data increases, using the HOG algorithm cannot effectively improve the recognition rate, because the traditional method has no dependence on the amount of data in the dataset.

**Table 3.** ResNet and HOG feature fusion recognition rate.

| Methods | Recognition Rate (%) | | | | | | | | | | | |
| | SDUST | | | | FYO | | | | NCUT | | | |
| | Twenty | Thirty | Forty | Fifty | Twenty | Thirty | Forty | Fifty | Twenty | Thirty | Forty | Fifty |
| HOG | 83.06 | 83.13 | 83.15 | 83.16 | 82.01 | 82.04 | 82.08 | 82.10 | 81.30 | 81.32 | 81.34 | 81.35 |
| ResNet | 84.97 | 90.47 | 90.86 | 92.30 | 86.60 | 89.59 | 91.93 | 93.43 | 83.93 | 87.70 | 89.13 | 90.06 |
| ResNet_HOG (ours) | **86.57** | **91.03** | **92.70** | **93.27** | **90.46** | **92.40** | **94.60** | **95.36** | **86.53** | **90.10** | **91.67** | **93.40** |

In addition, it can be seen from the table that the recognition rate of our proposed feature fusion method is better than that of using ResNet and HOG alone, which proves the feasibility of our proposed feature fusion method.

The fusion dataset is consistent with the experiments performed on the single dataset, and the experimental results are shown in Table 4. Table 4 shows the comparison between the proposed feature fusion method and the ResNet method, and it can be seen that the recognition rate of the proposed method is better than that of using ResNet and HOG alone.

**Table 4.** Recognition rate on the fused dataset after feature fusion.

| Methods | Recognition Rate (%) | | | |
| | Twenty | Thirty | Forty | Fifty |
|---|---|---|---|---|
| HOG | 67.34 | 67.36 | 67.37 | 67.39 |
| ResNet | 83.70 | 86.83 | 90.46 | 91.16 |
| ResNet_HOG (ours) | **85.70** | **89.46** | **92.27** | **93.47** |

*3.2. Feature Fusion Robustness Experiments*

The robustness of the model has always been the focus of cross-database dorsal vein recognition research, and traditional methods are not robust to cross-database dorsal vein images and datasets with Gaussian noise added. Here we conduct experiments on three different datasets using the HOG algorithm. In addition, we use the Partition Local Binary Patterns (PLBP) [24] algorithm for comparison. PLBP is an improvement based on the LBP algorithm. It divides an image into non-overlapping blocks, uses the LBP algorithm for each block, and finally splices the LBP feature statistical histograms of the entire image. The experimental results are shown in Table 5.

**Table 5.** Comparison of different methods.

| Methods | Recognition Rate (%) | | | |
| | SDUST | FYO | NCUT | Fusion Dataset |
|---|---|---|---|---|
| PLBP | 60.09 | 55.50 | 70.19 | 61.07 |
| HOG | 83.16 | 82.10 | 81.35 | 67.39 |
| ResNet | 92.30 | 93.43 | 90.06 | 91.16 |
| ResNet_HOG (ours) | **93.27** | **95.36** | **93.40** | **93.47** |

It can be seen from Table 5 that the recognition rate of using the PLBP and HOG algorithms alone not only does not achieve good results but also is much lower than the experimental results of other researchers [10] on the NCUT dataset. The reason for the analysis is that when other researchers conducted experiments on the NCUT dataset, they used the original dataset and did not use Gaussian noise to expand the dataset. Additionally, our experiments are performed on the dataset augmented with Gaussian noise, which also shows that traditional features are not robust to our dataset with Gaussian

noise added. Furthermore, it can be seen from Table 5 that in the FYO dataset, the effect of using the PLBP algorithm is particularly low, and we find some categories with the worst recognition results in the FYO dataset, as shown in Figure 10. Most of the categories with poor recognition results are recognized in the category 52. Figure 11 shows a statistical histogram of the texture information extracted from the DHV images using LBP with rotationally invariant consistency pattern. As can be seen from the figure, the texture information for categories 131 and 170 differs significantly from the registered features but differs very little from the registered features for category 52, which leads to DHV images like 131 and 170 being easily identified as the category 52. Analysis of the reasons for this occurrence, by adding Gaussian noise leads to a variation in the intra-class images, where the intra-class distances become larger and are thus misidentified as other classes.

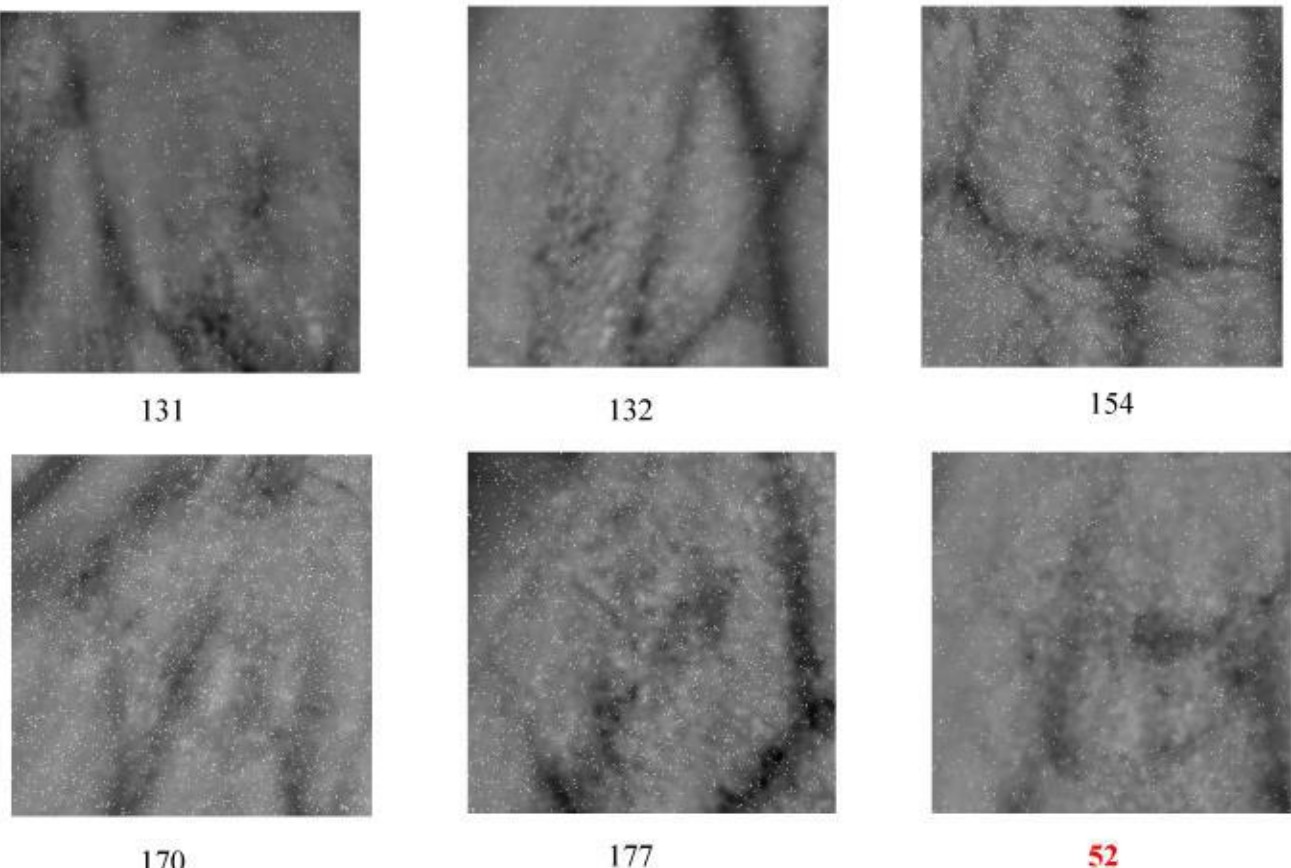

**Figure 10.** The most misclassified tags in the FYO dataset.

The HOG algorithm can achieve a recognition rate of more than 80% on a single database, but only 67.39% on a fusion dataset. We find some of the worst-recognized classes in the fused dataset, which are mostly data from NCUT and SDUST, as shown in Figure 12. Most of these partially identified worst classes are identified as 316 and 8, which are images in the FYO dataset, as shown in Figure 13.

We found that in the worst-recognized category, the blood vessel information of these images is not obvious, and the Gaussian noise on the images accounts for more. The images of the most misclassified categories have almost no blood vessel information, and most of the information is the hair on the back of the hand. The gap between these two categories cannot be seen visually. We visualize the HOG feature maps of these categories, as shown in Figure 14.

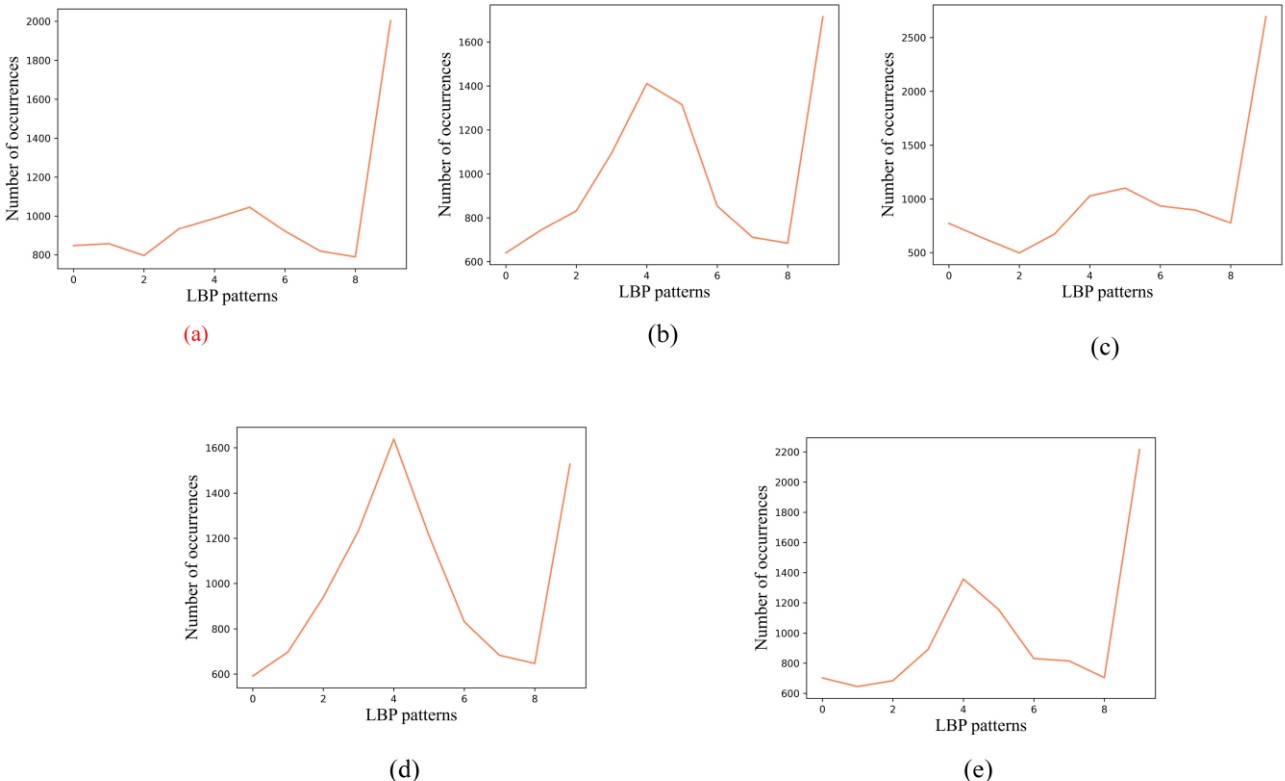

**Figure 11.** The misclassification of most categories of texture information. (**a**) Registration characteristics of category 52. (**b**) Registration characteristics of category 131. (**c**) Testing characteristics of category 131. (**d**) Registration characteristics of category 170. (**e**) Testing characteristics of category 170.

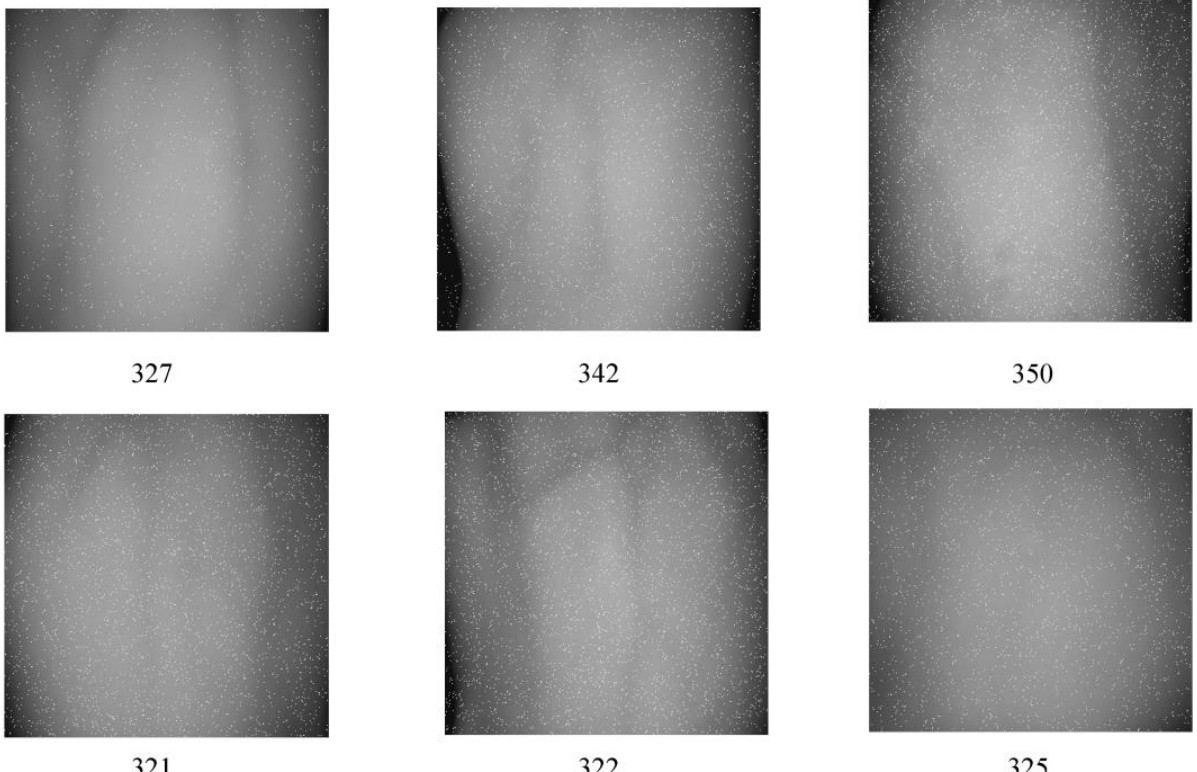

**Figure 12.** Fusion dataset part identifies the worst class.

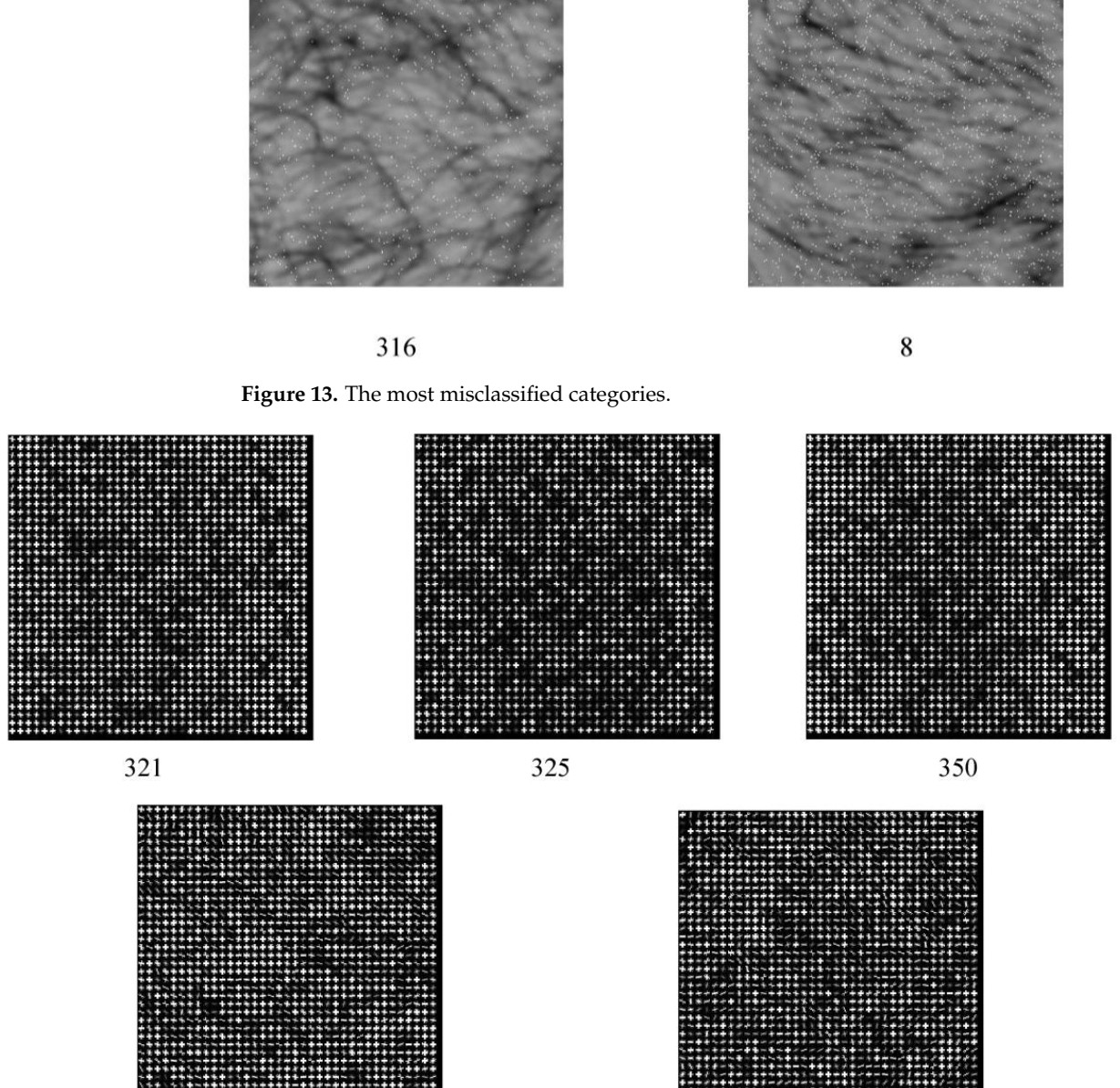

**Figure 13.** The most misclassified categories.

**Figure 14.** HOG feature visualization for misclassification.

As can be seen from Figure 14, the images with insignificant vein information in the NCUT and SDUST datasets and the images of these two categories in the FYO dataset have less obvious HOG features. Therefore, errors are prone to occuring when calculating the Euclidean distance between two categories, resulting in a low recognition rate using the HOG algorithm on the fusion dataset.

The recognition rate of the ResNet_HOG method proposed in this paper is significantly higher than that of using ResNet and HOG alone on a single dataset. Moreover, we also achieved good recognition rates on the fusion dataset and have strong robustness.

### 3.3. Comparison with Other Researchers

Recently, some researchers [25] achieved the current optimal results on the NCUT dataset using CNN and PLBP feature fusion. We adopted this idea and fused PLBP with ResNet for feature fusion, and the experimental results are shown in Table 6. As can be

seen from Table 6, our proposed method achieves the current optimal results on the SDUST dataset, but not on the FYO and NCUT. In [18], the authors used a decision-level fusion of palm, dorsal, and wrist biometric features on vein images, which can make full use of hand biometric features, so this method is superior to our proposed method. In [26], the authors used the principal component analysis (PCA) method to expand the DHV dataset to 250 per category, which far exceeded our data volume, so their experimental results were superior to ours. However, when we expand our dataset to 250 per category, the recognition rate is 99.93%, and the recognition results are superior to [26].

**Table 6.** Recognition rates of different feature fusion methods on a single database.

| Methods | Recognition Rate (%) | | | | | | | | | | | |
| --- | --- | --- | --- | --- | --- | --- | --- | --- | --- | --- | --- | --- |
| | SDUST | | | | FYO | | | | NCUT | | | |
| | Twenty | Thirty | Forty | Fifty | Twenty | Thirty | Forty | Fifty | Twenty | Thirty | Forty | Fifty |
| ResNet_PLBP | 85.40 | 90.87 | 91.77 | 92.76 | 90.27 | 91.77 | 94.17 | 95.20 | 85.96 | 89.73 | 90.17 | 92.50 |
| VeinNet [17] | 92.28 | | | | \ | | | | \ | | | |
| Skeleton [27] | \ | | | | \ | | | | 92.75 | | | |
| CNN Model [18] | \ | | | | 98.90 | | | | \ | | | |
| CNN [26] | \ | | | | \ | | | | 99.61 | | | |
| ResNet_HOG (ours) | **86.57** | **91.03** | **92.70** | **93.27** | **90.46** | **92.40** | **94.60** | **95.36** | **86.53** | **90.10** | **91.67** | **93.40** |

Table 7 shows the comparison between our proposed feature fusion method and the current methods of cross-database, from which it can be seen that our method can achieve better results, except for [16]. The reason is that [15,16,28] use datasets collected through different devices and the same subjects, whereas we use datasets with different devices, different subjects, and different ethnicities, and these different factors have a significant impact on DHV identification [29], so our method is slightly below [16].

**Table 7.** Recognition rates of different feature fusion methods on fusion dataset.

| Methods | Recognition Rate (%) | | | |
| --- | --- | --- | --- | --- |
| | Twenty | Thirty | Forty | Fifty |
| ResNet_PLBP | 84.47 | 89.26 | 92.13 | 92.63 |
| Improved SIFT [15] | 88.50 | | | |
| SIFT [28] | 90.17 | | | |
| Two-stage Coarse-to-fine Matching [16] | 96.80 | | | |
| ResNet_HOG (ours) | **85.70** | **89.46** | **92.27** | **93.47** |

*3.4. Comparison between Feature Fusion and Data Volume*

Through the experiments in the previous sections, we can see that increasing the number of samples in the dataset can improve the recognition rate of the dorsal vein of hand, but of the two methods, feature fusion achieved better results in the recognition of dorsal vein in small samples. Figure 15 shows the relationship between feature fusion and the number of samples.

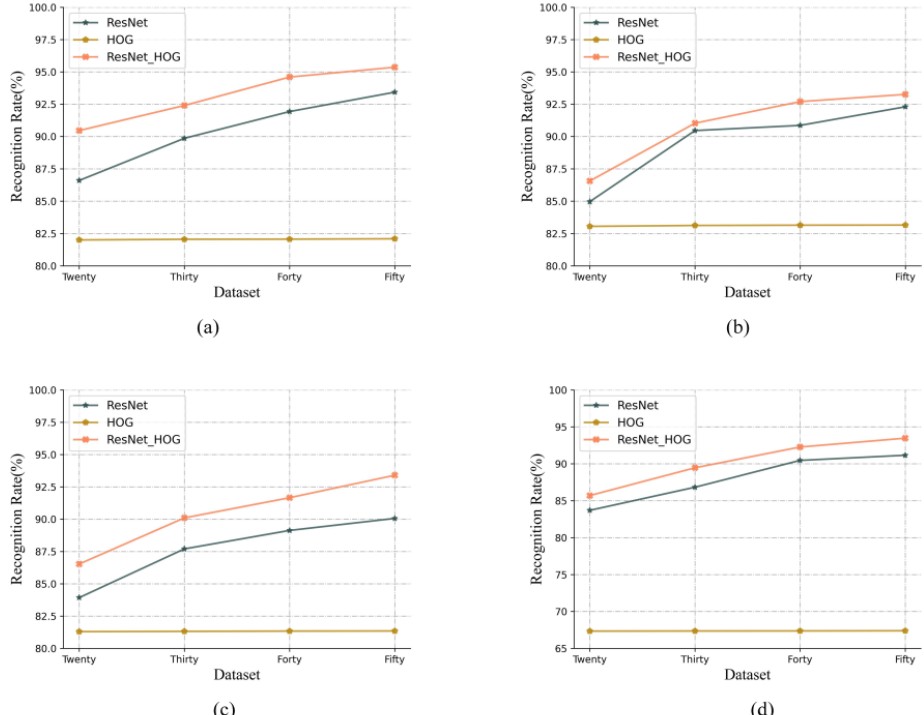

**Figure 15.** Data volume vs. feature fusion. (**a**) Results of ResNet, HOG, and ResNet_HOG on FYO dataset. (**b**) Results of ResNet, HOG, and ResNet_HOG on SDUST dataset. (**c**) Results of ResNet, HOG, and ResNet_HOG on NCUT dataset. (**d**) Results of ResNet, HOG, and ResNet_HOG on Fusion dataset.

From Figure 15, we can first see that our proposed feature fusion method achieved good results on the small sample DHV dataset. Secondly, we can see that when the amount of data reaches 40 pieces per category, the proposed feature fusion method exceeds the ResNet recognition rate of 50 pieces without feature fusion. This is because the shallow semantic information of the image is extracted using ResNet and then fused with HOG features; after a series of convolution operations, the final features for classification include both the deep semantic information and the gradient information of the image. Through such a feature fusion method, the features of the image can be fully obtained and thus be accurately classified.

## 4. Discusion

### 4.1. Gamma Value Influence

When a picture appears too bright or too dark, it leads to poor image contrast, and that is when Gamma correction needs to be performed. Our dataset was grayed out for the experiments, so the color information did not affect the experiments much. The Gamma correction of the images is required to make the black areas of the images appear brighter. When Gamma < 1, in the high gray value area, the dynamic range becomes smaller, the image contrast decreases, the overall gray value of the image becomes larger, and the image becomes brighter. When Gamma > 1, gamma >1 in the low gray value area, the dynamic range becomes smaller, the image contrast decreases, and the overall gray value of the image becomes smaller and darker. To find the appropriate value for Gamma < 1, we conducted four sets of experiments on three different datasets, and the experimental results are shown in the Table 8. From the table, we can see that the value of Gamma does affect different datasets, but overall, the effect of Gamma on the experimental results is minimal, and in most of the literature, Gamma is generally taken as 0.5, so in this paper, we also take 0.5.

**Table 8.** The influence of different gamma values on the experiment.

| Dataset | Gamma Value | | | | |
|---|---|---|---|---|---|
| | **0.2** | **0.4** | **0.5 (ours)** | **0.6** | **0.8** |
| NCUT | 86.52 | 86.61 | 86.53 | 86.58 | 86.60 |
| FYO | 90.26 | 90.35 | 90.46 | 90.38 | 90.42 |
| SDUST | 86.59 | 86.51 | 86.57 | 86.44 | 86.49 |

*4.2. Influence of Gaussian Noise Intensity Model*

How the SNR affects the recognition rate of ResNet, we have carried out three sets of experiments on the Twenty dataset of NCUT, as shown in Table 9. In the table, we can see that the recognition rate of ResNet gets lower and lower as the noise increases. This is mainly because, when adding too much Gaussian noise, the vein information on the image is completely covered by the noise, which makes the model difficult to train and the recognition rate decreases.

**Table 9.** Influence of Gaussian noise intensity model.

| Noise Range | 0.1–0.3 | 0.3–0.5 | 0.5–0.7 |
|---|---|---|---|
| Recognition Rate (%) | 77.45 | 72.94 | 66.81 |

*4.3. Influence of Convolution Blocks*

We performed a separate convolution operation before inputting the image and HOG features into the ResNet residual structure, and this operation had an effect on the experimental results, which we performed on three different datasets, and the experimental results are shown in Table 10. From the Table 10, we can see that adding the convolutional block performs significantly on the NCUT and SDUST datasets but not on the FYO dataset, although the recognition rate with the convolutional block is better than that without the convolutional block. This is because without adding the convolutional block, reshaping the HOG feature size will lead to the loss of most of the HOG features, which will affect the recognition results.

**Table 10.** Experimental comparison with and without convolution blocks.

| Methods | Recognition Rate (%) | | |
|---|---|---|---|
| | **NCUT** | **FYO** | **SDUST** |
| With convolution blocks | 86.53 | 90.46 | 86.57 |
| No convolution blocks | 83.60 | 90.23 | 85.74 |

*4.4. The Influence of Increasing the Amount of Data HOG on the Model*

Through the experiments in Section 3, we can see that ResNet_HOG outperforms ResNet alone for small sample dorsal hand vein recognition, but the gain obtained using ResNet_HOG always seems to be around 1–3% regardless of the dataset size, and we analyze the reasons for this as follows.

In this paper, traditional features play an auxiliary role. Traditional features can extract information that cannot be obtained by depth features (such as gradient information, etc.), but this information has a limited impact on depth features, so the recognition results of the model are not significantly improved after performing feature fusion. We also found this problem in [17], where the authors used a fusion of ResNet and LBP features and only achieved a 1.97% higher recognition rate than using ResNet alone. In addition, we conducted four sets of experiments on the NCUT dataset to determine how much HOG affects ResNet. As can be seen from the experiments in the Table 11, the effect of feature

fusion seems to become less and less pronounced as the amount of data increases. This is because as the amount of data increases, ResNet has enough samples for training and the trained model becomes more and more robust, so it is possible to obtain a high recognition rate using only ResNet.

**Table 11.** Experimental comparison of different data volumes.

| Methods | Recognition Rate (%) | | | |
|---|---|---|---|---|
| | **One Hundred Samples** | **One Hundred and Fifty Samples** | **Two Hundred Samples** | **Two Hundred and Fifty Samples** |
| ResNet | 96.93 | 98.07 | 98.92 | 99.54 |
| ResNet_HOG | 97.89 | 98.84 | 99.27 | 99.93 |

For the above analysis, we can see that either ResNet and HOG feature fusion or ResNet with LBP for feature fusion can outperform the recognition rate obtained using ResNet alone. However, because the recognition rate obtained using ResNet alone is more than 90%, the model's recognition rate is only improved by 1–3% after feature fusion, which is also effective for a dataset with few samples.

*4.5. Other Deep Learning vs. Traditional Methods Discussion*

In [17], the authors conducted experiments using the ResNet network. The authors performed two types of data enhancement (increasing the amount of data and changing the image brightness), and the recognition rate obtained by both enhancement methods was better than that obtained by using ResNet alone. In addition, the authors also performed the method of ResNet and LBP feature fusion, and the recognition rate after performing data enhancement using ResNet was 90.31%, while the recognition rate after performing ResNet and LBP feature fusion was 92.28%, which is an improvement but not significant. Our analysis shows that the black background information of the rotated image has an impact on the extraction of LBP features, which leads to the insignificant improvement of recognition results after feature fusion.

We searched many references and found no literature combining DL and HOG, but there are experiments with CNN and PLBP feature fusion. In [25], the authors designed three methods of CNN and PLBP feature fusion, namely serial fusion, decision fusion, and feature fusion. The decision fusion and feature fusion are the best, which are 0.34% higher than the CNN network without fusion.

According to the preceding literature, traditional features and deep learning for feature fusion not only excel in a few sample dorsal hand vein recognition but also improve the recognition rate of large sample data, demonstrating the feasibility of traditional features and deep learning for feature fusion.

## 5. Conclusions

In this paper, we design a method for the fusion of ResNet and HOG features, which achieves better results on small sample datasets. We adopt the methods of other researchers and conduct experiments on our dataset, and the experimental results show that the recognition rate of the feature fusion method is better than that of using ResNet alone. This proves that the combination of deep learning and traditional features can not only solve the problem that deep learning has a low recognition rate for small samples but also solve the problem that traditional features are not robust to Gaussian noise. It further illustrates the superiority and feasibility of using deep learning and traditional feature fusion in the field of DHV recognition.

At present, our work has achieved good results on the dataset with Gaussian noise. In the future, we will utilize more ways to expand the dataset for verification, such as physical expansion (random rotation, image translation, image exposure, etc.) and deep learning automatically expansion [30,31]. Our fusion dataset now includes three different datasets,

and we hope to obtain more datasets in the future to expand the database of DHVs. In addition, to explore the possibility of feature fusion between deep features and traditional features, we will use a variety of traditional features and deep feature fusion methods to verify the DHV dataset.

**Author Contributions:** Investigation, J.L.; data curation, K.L. (Keming Li), J.W. and Y.Y.; software, K.L. (Kefeng Li) and G.Z.; writing—original draft preparation, J.L. and K.L. (Kefeng Li); writing— review and editing, K.L. (Kefeng Li). All authors have read and agreed to the published version of the manuscript.

**Funding:** This research received no external funding.

**Institutional Review Board Statement:** The study was conducted in accordance with the Declaration of Helsinki, and approved by the The Ethics Committee of Shandong Jiaotong University(protocol code 3701063670893 and date of approval 27 July 2022).

**Informed Consent Statement:** Informed consent was obtained from all subjects involved in the study. Written informed consent has been obtained from the participants to publish this paper.

**Data Availability Statement:** The datasets used in this paper are all available on request from the authors of the following three articles. (1) **SDUST:** [17]. (2) **FYO:** [18]. (3) **NCUT:** [11].

**Conflicts of Interest:** The authors declare no conflict of interest.

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
