# Peer review of "Recognition of Dorsal Hand Vein in Small-Scale Sample Database Based on Fusion of ResNet and HOG Feature"

_electronics, doi:10.3390/electronics11172698_

Round 1

Reviewer 1 Report

The article addresses the interesting topic of automatic recognition of persons identity through images of the dorsal hand vein. In particular, the authors focus on the combination of traditional image processing techniques with deep learning methods. They achieve two main results: a new dataset for DHV recognition built from the fusion of three different databases, and a novel architecture that uses both HOG features and features learned by a ResNet. I recognise the relevance of the paper and its interest for the research in this field, however I would ask the authors to clarify and better explain some choices before publication: 

1) Are images normalised (e.g. z-score) once "fused" into a single dataset? I think some form of normalisation might be helpful since images are collected with different devices; 

2) Why is Gaussian noise added to all images (if it is so)? I also do not understand whether this is done to augment the dataset (i.e. increase the training sample size) or not. If so, I would recommend other types of augmentation techniques such as simple symmetry transformations (e.g. rotations); 

3) The architecture is made of a convolutional block taking the image as input and the HOG features which are also fed into a different convolutional block, in this way the feature maps are combined and then pass through the ResNet. How are those convolutional block trained? Is it an end-to-end process (i.e. the only loss is the one in the final FC of the ResNet)? 

4) Is it really necessary to have these two convolutional blocks before the ResNet? Wouldn't be sufficient to reshape and resize the HOG features and then fed everything directly into the ResNet? Especially, the convolutional block that takes as input the image seems superfluous. 

5) It seems that the gain achieved by the ResNet+HOG is little compared to the results of the ResNet alone. Can the authors comment on this? 

6) I would expect the HOG features to help the model when the dataset is limited in size. However, it seems the gain obtained with ResNet+HOG is always around 1-3 % for all datasets regardless of the size. Can the authors comment on this? 

7) The comparison with other papers and in general the discussion of the relevant literature needs to be improved. For instance, are there other papers using ResNet for DHV? What about other references combining DL and other feature extraction methods (such as HOG)? Have previous results been improved somehow (even if I understand that the model is tested on a "new" dataset, still some comparison can be made)? 

Finally, there is a typo "Error! Reference source not found." in the section where the three initial datasets are introduced and it must be corrected. If the authors are able to answer the above questions and improve the text of the manuscript accordingly, then I would be glad to consider the paper for publication. 

Reviewer 2 Report

1.       Improve the abstract. The abstract needs to highlight the contributions of the work.

2.       Several acronyms are not defined, or the definitions appears not in first mention. HOG is not well defined.

3.       The paper has grammar issues that make redundant or hard to understand. In line 158 “formula” has an extra “e”. Check lines 20,174,229,230.

4.       There are errors in references in Section 2.

5.       The caption in Figure is ambiguous and in the image is blurry.

6.       There is an error in Equation (1).

7.       In [21] a static square size was chosen to separate the vein pattern from the background. This method works since only a database were used and all the images were taken using the same configurations. In this work different datasets are used, and the background differs due to sensors, angles, light conditions, distance, etc. For that reason, in Figure 3a and 3c the background is still visible, and in Figure 3b the vein pattern is not complete. Is better to use a segmentation algorithm to correctly select the region of interest.

8.       What is the SNR used in this experiment? How this parameter affects the recognition rate of ResNet and the proposed method? Why not using other artificial expand of dataset technics such as SMOTE?

9.       The parameter gamma = 0.5 is static and valid for all the datasets? Why? If not, how do you choose the optimal gamma?

10.   Figure 15 is blurry.

Round 2

Reviewer 2 Report

The authors attended all my suggestions and clarify my doubts. 

recommend this paper for publication.